# Novel Synergistic Anti-Enteroviral Drug Combinations

**DOI:** 10.3390/v14091866

**Published:** 2022-08-25

**Authors:** Aleksandr Ianevski, Eva Zusinaite, Tanel Tenson, Valentyn Oksenych, Wei Wang, Jan Egil Afset, Magnar Bjørås, Denis E. Kainov

**Affiliations:** 1Department of Clinical and Molecular Medicine (IKOM), Norwegian University of Science and Technology, 7028 Trondheim, Norway; 2Institute of Technology, University of Tartu, 50411 Tartu, Estonia; 3Department of Medical Microbiology, St. Olavs Hospital, 7028 Trondheim, Norway; 4Institute for Molecular Medicine Finland, University of Helsinki, 00014 Helsinki, Finland

**Keywords:** echovirus, enterovirus, broad-spectrum antiviral agent, antiviral drug combination, antiviral strategy

## Abstract

*Background:* Enterovirus infections affect people around the world, causing a range of illnesses, from mild fevers to severe, potentially fatal conditions. There are no approved treatments for enterovirus infections. *Methods:* We have tested our library of broad-spectrum antiviral agents (BSAs) against echovirus 1 (EV1) in human adenocarcinoma alveolar basal epithelial A549 cells. We also tested combinations of the most active compounds against EV1 in A549 and human immortalized retinal pigment epithelium RPE cells. *Results:* We confirmed anti-enteroviral activities of pleconaril, rupintrivir, cycloheximide, vemurafenib, remdesivir, emetine, and anisomycin and identified novel synergistic rupintrivir–vemurafenib, vemurafenib–pleconaril and rupintrivir–pleconaril combinations against EV1 infection. *Conclusions:* Because rupintrivir, vemurafenib, and pleconaril require lower concentrations to inhibit enterovirus replication in vitro when combined, their cocktails may have fewer side effects in vivo and, therefore, should be further explored in preclinical and clinical trials against EV1 and other enterovirus infections.

## 1. Introduction

Enteroviruses affect people around the world, causing the common cold, hand-foot-and-mouth disease, meningitis, myocarditis, pancreatitis, and poliomyelitis. They are also associated with chronic diseases such as type 1 diabetes, asthma, and allergies. Enteroviruses are non-enveloped, positive-sense, single-stranded RNA viruses. They belong to the Enterovirus genus of the *Piconaviridae* family. The genus includes 12 species (enteroviruses A–H, and J and rhinoviruses A–C) which include echovirus 6 (EV6), poliovirus, coxsackievirus, enterovirus D68, enterovirus A71, and rhinovirus. While poliovirus has been largely eradicated worldwide with successful vaccination, non-polio enteroviruses continue to emerge.

There is a lack of approved antiviral drugs that can be deployed to treat enterovirus infections [1]. Several capsid-binding agents (such as pleconaril) as well as 3C protease inhibitors (such as rupintrivir) failed in several clinical trials of human enterovirus infection due to limited efficacy or side effects [2,3,4,5]. In recent years, some progress has been made in developing novel anti-enteroviral drug candidates with both in vitro and in vivo antiviral efficacy [1]. However, there are still several hurdles that need to be overcome before moving the drug candidates to human clinical trials. First, there is a need for broad-spectrum antiviral agents (BSAs) inhibiting EV6, EV-A71, CV-A16, CV-A10, EV-D68, rhinoviruses, and other members of the enterovirus family. Second, there is a need for antivirals with a high genetic barrier to drug resistance. Resistant mutants have been described for almost all direct-targeting antivirals in cell culture, raising the concern that resistance might quickly emerge when antivirals are advanced to the market [1]. Third, as the susceptible populations of enteroviruses are mainly children and infants, and the infection is generally not lethal, the BSAs should be extremely safe.

To target multiple enteroviruses and mitigate the development of antiviral drug resistance, antivirals are combined into drug cocktails [6,7]. Our recent studies have revealed synergism of several BSA-containing drug combinations (BCCs), including vemurafenib plus homoharringtonine, gemcitabine, obatoclax or rupintrivir against EV1 [8]. Importantly, these synergistic drug cocktails contain lower concentrations of antivirals, which could decrease side effects of individual drugs. However, new synergistic drug combinations need to be identified to inhibit different enteroviruses.

Here, we identified novel BCCs in vitro. In the long term, we expect that these BCCs will benefit patients suffering from enterovirus-mediated diseases, improving disease outcomes and reducing hospitalization time and treatment costs.

## 2. Materials and Methods

### 2.1. Drugs

Appendix A lists compounds, their suppliers, and catalogue numbers. To obtain 10 mM stock solutions, compounds were dissolved in dimethyl sulfoxide (DMSO; Sigma-Aldrich, Darmstadt, Germany) or milli-Q water. The solutions were stored at -80 °C until use.

### 2.2. Cell Cultures

RPE cells were grown in DMEM-F12 supplemented with 10% FBS (fetal bovine serum), 100 μg/mL streptomycin, and 100 U/mL penicillin (Pen-Strep). Human adenocarcinoma alveolar basal epithelial A549 cells were grown in DMEM supplemented with 10% FBS and Pen-Strep. The cell lines were maintained at 37 °C with 5% CO_2_ in a humidified atmosphere.

### 2.3. Viruses

EV1 (Farouk strain; ATCC) was provided by Prof. Marjomäki from University of Jyväskylä. EV1 was amplified in a monolayer of A549 cells in the DMEM media containing Pen/Strep and 0.2% BSA (bovine serum albumin). Virus stocks were stored at −80 °C.

### 2.4. Drug Test

Approximately 4 × 10^4^ A549, or RPE cells were seeded per well in 96-well plates. The cells were grown for 24 h in DMEM or DMEM-F12 supplemented with 10% FBS and Pen-Strep. The medium was replaced with DMEM or DMEM-F12 containing 0.2% BSA and Pen-Strep. The compounds were added to the cells in 3-fold dilutions at 7 different concentrations, starting from 30 μM. No compounds were added to the control wells. The cells were mock- or virus-infected at an moi (multiplicity of infection) of 0.1. After 48 (A549 cells) or 72 (RPE cells) hours of infection, the medium was removed from the cells, and a CellTiter-Glo assay (Promega) was performed to measure cell viability.

The half-maximal effective concentrations (EC_50_) were calculated using drugvirus.info server [9], based on the analysis of the viability of infected cells by fitting drug dose–response curves using a four-parameter (4PL) logistic function *f*(*x*):(1)$$f\left(x\right)=A_{min}+\frac{A_{max}-A_{min}}{1+{\left(\frac{x}{m}\right)}^{\lambda }},\;$$
where *f*(*x*) is a response value at dose *x*, *A_min_* and *A_max_* are the upper and lower asymptotes (minimal and maximal drug effects), *m* is the dose that produces the half-maximal effect (EC_50_ or CC_50_), and *λ* is the steepness (slope) of the curve. The relative effectiveness of the drug was defined as the selectivity index (SI = CC_50_/EC_50_).

To quantify each drug response in a single metric, a drug sensitivity score (DSS) was calculated as a normalized version of the standard area under dose–response curve (AUC), with the baseline noise subtracted, and the normalized maximal response at the highest concentration (often corresponding to off-target toxicity):(2)$$\mathrm{DSS}=\frac{\mathrm{AUC}-t\left(x_{max}-x_{min}\right)}{\left(100-t\right)\left(x_{max}-x_{min}\right)\log_{10}A_{min}},\;$$
where activity threshold *t* equals 10%, and DSS is in the 0–50 range [10,11].

### 2.5. Drug Combination Test and Synergy Calculations

A549 or RPE cells were treated with different concentrations of drug pairs and infected with EV1 (moi 0.1) or mock. The viability of A549 and RPE cells was measured using CellTiter-Glo after 48 or 72 h of infection, respectively.

To test whether the drug combinations act synergistically, the observed responses were compared with expected combination responses. The expected responses were calculated based on the ZIP reference model using SynergyFinder version 3 [12]. Synergy scores were quantified as average excess response due to drug interactions (i.e., 10% of cell survival beyond the expected additivity between single drugs represents a synergy score of 10). Additionally, we calculated most synergistic area (MSA) scores for each drug combination, i.e., synergy scores calculated for the most synergistic 3-by-3 dose-windows in dose–response matrices. To eliminate cytotoxic synergistic effect, we first subtracted viability of mock-infected from virus-infected cells, which allowed us to calculate synergistic effect specific to EV1 inhibition.

### 2.6. Plaque Reduction Assay

EV1 titers were determined by plaque assay on A549 cells, as described earlier [8,13,14].

## 3. Results

Previously, our group has extensively reviewed and identified >250 approved, investigational, and experimental BSAs [7,9]. We made a database to summarize information on the BSAs (https://drugvirus.info/, accessed on 30 July 2022).

We have recently tested 45 BSAs against EV1 and/or EV6 in human cancerous lung epithelial A549 cells [8,13,14]. From this, we identified anti-enterovirus activities for cycloheximide, vemurafenib, digoxin, anisomycin, emetine, homoharringtonine, obatoclax, gemcitabine, and dalbavancin. Thus, we expanded the spectrum of anti-enteroviral activities for the BSAs. Other BSAs could also have anti-enterovirus activities.

Here, we tested 239 BSAs against EV1 in A549 cells. Eight different concentrations of the compounds were added to virus-infected cells. Cell viability was measured after 48 h to determine compound efficiency. After the initial screen, we identified pleconaril, nelfinavir, rupintrivir, vemurafenib, remdesivir, trametinib, emetine, cycloheximide, atovaquone, homoharringtonine, salinomycin and enoxacin as compounds that rescued virus-infected cells from death (DSS > 10; Appendix A). Some of the identified compounds possess a structure–activity relationship (Figure 1a).

We repeated the antiviral efficacy experiment as well as measured toxicity of hit compounds on A549 cells. We validated anti-enteroviral activities of 7 BSAs. Pleconaril, rupintrivir, cycloheximide, vemurafenib, remdesivir, emetine, and anisomycin showed high selectivity (SI > 100; Figure 1b,c). Viral capsid-binding pleconaril had the highest SI, followed by 3C protease inhibitor rupintrivir, translation inhibitor cycloheximide and virus-directed vemurafenib.

Rupintrivir has broader spectrum of antiviral activities than pleconaril [9]. However, both BSAs failed in some clinical trials of human enterovirus infection due to limited efficacy or side effects [2,3,4,5]. We hypothesized that rupintrivir efficacy could be increased and toxicity could be decreased by synergistically combining it with other anti-enteroviral BSAs.

We examined rupintrivir combinations with virus-directed vemurafenib, remdesivir, gemcitabine or pleconaril, as well as with host-directed anisomycin, cycloheximide, halofuginone, dalbavancin, emetine, homoharringtonine, obatoclax, or digoxin. Virus- and mock-infected A549 cells were treated with an increasing concentration of two drugs. After 48 h, cell viability was measured. We calculated the synergy scores for the most synergistic areas (MSA) of the drug interaction landscapes considering drug toxicity (Figure 2a–d). We found that rupintrivir plus vemurafenib, pleconaril, or cycloheximide were strongly synergistic (synergy scores > 10), whereas rupintrivir plus remdesivir, dalbavancin, anisomycin, and emetine were additive (synergy scores ranging from +5 to −5). Other combinations were antagonistic (synergy scores < −5, Figure 2d). Thus, rupintrivir plus vemurafenib, pleconaril, or cycloheximide protected cells from EV1-mediated death at lower concentration and more effectively than drugs alone.

Importantly, rupintrivir–vemurafenib, vemurafenib–pleconaril and rupintrivir–pleconaril combinations had the highest synergy scores in A549 cells (Figure 2d,e). These combinations were also synergistic in human nonmalignant RPE cells against EV1. All three combinations at selected concentrations reduced the EV1 production in comparison to drugs alone (Figure 2f). Thus, we identified synergistic combinations that could result in increased efficacy and decreased toxicity to inhibit EV1 infection in vitro.

## 4. Discussion

Here, we identified novel synergistic combinations of rupintrivir–vemurafenib, vemurafenib–pleconaril, rupintrivir–pleconaril and rupintrivir–cycloheximide against enterovirus EV1 infection. Given that vemurafenib prevents infection of some but not all enteroviruses (patent WO2020070390A1) [8], and cycloheximide has an irreversible effect on protein translation [15], the further development of rupintrivir–pleconaril combination could be prioritized.

It was shown that both rupintrivir and pleconaril inhibit replication of multiple enteroviruses [16]. The combination of these agents could mitigate the development of antiviral drug resistance and could lead to the development of pan-genus antiviral therapeutics. Low doses of BSAs in combination decreases their toxicity in vitro and, thus, could lower their side effects in vivo [8]. Analysis of the drug–target interactions, mechanisms of drug actions, their immunomodulatory properties, and routes of administration [7] indicates that further development of this combination could be prioritized. Translation of this drug combination could also save time and cost due to the developmental status of both rupintrivir and pleconaril.

Thus, our study establishes a resource and platform for future development of novel BCCs and unveil new insights into the antiviral research. Our study also increased the availability and accessibility of antiviral options, resulting in more antiviral treatments. Thus, our work has the potential to decrease morbidity and mortality, increase healthy life years, and improve the quality of life for infected individuals.

## 5. Conclusions

There are dozens of antiviral therapies in development. Many more are awaiting their discoveries. Here, we used high-throughput biology to identify promising anti-enterovirus drug combinations. We showed that synergistic combinations of the virus-directed rupintrivir and pleconaril require lower concentration of drugs than monotherapies to inhibit EV1 replication in vitro and, thus, could have fewer side effects in vivo. We also confirmed our previous hypothesis that the combination of virus-directed drugs could have more benefits than combinations of virus- and host-, or only host-directed antivirals [7].

## Figures and Tables

**Figure 1 viruses-14-01866-f001:**
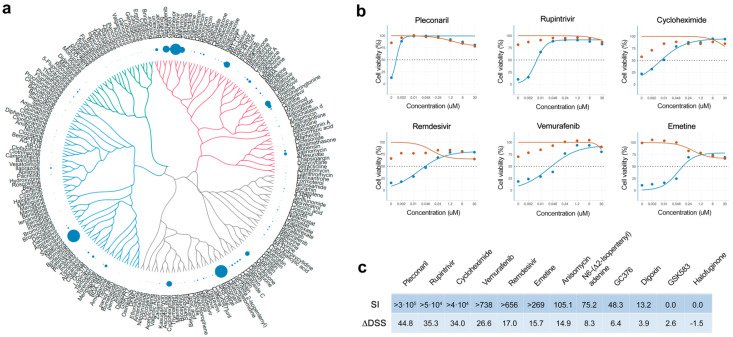
Anti-EV1 activity of 239 broad-spectrum antivirals (BSAs) in A549 cells. (**a**) Structure–antiviral activity relation of BSAs. A549 cells were treated with increasing concentrations of a compound and infected with the EV1 (moi, 0.1). After 48 h, the viability of the cells was determined using the CellTiter-Glo (CTG) assay. The anti-EV1 activity of the compounds was quantified using the drug sensitivity scores (DSS) and shown as bubbles. Bubble size corresponds to compounds DSSs. The compounds were clustered based on their structural similarity calculated by ECPF4 fingerprints and visualized using the D3 JavaScript library. (**b**) Validation of anti-EV1 activity of hit compounds in A549 cells. A549 cells were treated with increasing concentrations of a compound and infected with the EV1 (moi, 0.1: blue) or mock (red). After 48 h, the viability of the cells was determined using the CTG assay. Mean ± SD; n = 3. Plots for 6 most effective BSAs are shown. (**c**) Table showing selectivity indexes (SI = CC50/EC50) and ΔDSS for selected BSAs.

**Figure 2 viruses-14-01866-f002:**
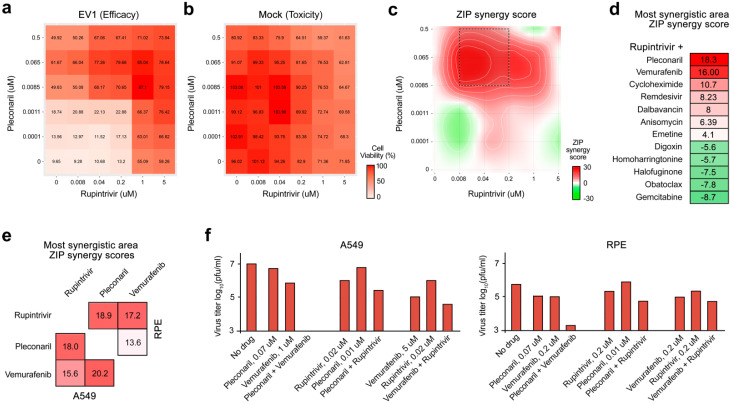
Anti-EV1 activity of BSA-containing combinations (BCCs). (**a**) The interaction landscape of the rupintrivir–pleconaril combination was measured using the CellTiter-Glo (CTG) assay on EV1-infected cells. (**b**) The interaction landscape of the rupintrivir–pleconaril combination measured using the CTG assay on mock-infected cells. (**c**) Synergy interaction landscape for selectivity (Selectivity=Efficacy-(100-Toxicity)) of the drug combination. (**d**) ZIP synergy scores calculated for the most synergistic areas (MSA) of interaction landscapes for selectivity obtained for rupintrivir-containing drug combinations on A549 cells. (**e**) Heatmap showing ZIP synergy scores calculated for the most synergistic areas (MSA) of interaction landscapes for selectivity obtained for 3 BSA combinations on A549 and RPE cells. (**f**) The effects of different concentrations of rupintrivir, pleconaril, vemurafenib and their combinations on replication of EV1 in A549 and RPE cells measured by plaque reduction assay.

## Data Availability

All data generated or analyzed during this study are included in this published article and its Appendix A.

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
