# Peer review of "Novel Synergistic Anti-Enteroviral Drug Combinations"

_viruses, 2022, doi:10.3390/v14091866_

Round 1

Reviewer 1 Report

This paper by Ianevski et al. proposed effective drug combinations against enterovirus. The author firstly screened the libraries of broad-spectrum antiviral agents using A549 cells and selected drug candidates with high antiviral activity. Then drug combinations against EV1 infection were further identified in vitro using both A549 and RPE cells, which inhibited EV1 replication with lower dosage combination. The combinations of BSAs in cocktails also showed inhibitory effects on different members of the enterovirus family. Efficacious drug combination proved in vitro might show better efficacy and less toxicity in vivo. The paper is very well written, and proposed new synergistic combinations containing broad-spectrum antivirals drugs, which enables further studies in pre-clinical and clinical trials in the future. Although this identification of novel antiviral drug combinations was very interesting, there are some problems, which must be solved before it is considered for publication. If the following problems are well-addressed, this reviewer believe that the essential contribution of this paper are important for antiviral treatment.

 General comments:

1. There are several ways to determine the synergy score. In this paper, the author only chose ZIP to do the synergy calculation. The reviewer wondered if this is the best way to determine the synergistic effect of drug combinations. Would it be better to evaluate the drug response in a set of appropriate methods instead of just one?

2.  In this study, the efficacy of drug combinations was evaluated using only echovirus 1 (EV1). There was no information about other members of the enterovirus family. The reviewer believe it’s not persuasive enough about the claim that the novel synergistic drug combination inhibit different enteroviruses in the conclusion.

3. In figure 1a, the author mentioned there was a structure-activity relationship in some identified drugs. Is drug with certain structure properties prone to work better with other ones with given structure properties in antiviral treatment?

Author Response

  1. There are several ways to determine the synergy score. In this paper, the author only chose ZIP to do the synergy calculation. The reviewer wondered if this is the best way to determine the synergistic effect of drug combinations. Would it be better to evaluate the drug response in a set of appropriate methods instead of just one?

Re: We agree with the reviewer that there are no specific guidelines for choosing a synergy model. In addition, the exact mechanisms of action of many drugs are not well defined, making it difficult to make definite assumptions about the choice of a synergistic model. SynergyFinder combines the most popular synergy models - Bliss and Loewe, creating the ZIP synergy evaluation method. Therefore, the authors used the SynergyFinder (ZIP method) for synergy analysis.

  1. In this study, the efficacy of drug combinations was evaluated using only echovirus 1 (EV1). There was no information about other members of the enterovirus family. The reviewer believe it’s not persuasive enough about the claim that the novel synergistic drug combination inhibit different enteroviruses in the conclusion.

Re: We agree with the reviewer that the efficacy of drug combinations has not been tested against other enteroviruses. However, rupintrivir, pleconaril and vemurafenib have been shown to inhibit the replication of many enteroviruses. Based on this, we assumed that new combinations of these drugs can inhibit various enteroviruses. We modified the conclusion accordingly: “Conclusions: Because rupintrivir, vemurafenib, and pleconaril require lower concentrations to inhibit enterovirus replication in vitro when combined, their cocktails may have fewer side effects in vivo and therefore should be further explored in preclinical and clinical trials. against EV1 and other enterovirus infections".

  1. In figure 1a, the author mentioned there was a structure-activity relationship in some identified drugs. Is drug with certain structure properties prone to work better with other ones with given structure properties in antiviral treatment?

Re: Indeed, some structurally related drugs (including emetine and homoharringtonine; vemurafenib, rupintivir, maraviroc) are active against EV1. However, we do not yet know if a drug with a particular structural property works synergistically with another drug with a different structural property in antiviral treatment. To resolve this issue, it is necessary to expand the BSA library (now it is limited to 255 antiviruses) and test it for many different viruses (PMID: 33080984; 35402870; 35610052). We will try to answer this interesting question in our future antivirus research.

Reviewer 2 Report

The manuscript of Ianewski and colleagues “Novel synergistic anti-enteroviral drug combinations” identifies combinations of specific antivirals and broad-range antivirals together possessing synergistic properties and therefore of highly prospective for further use in treatmentpractice. In the course of the study confirmed were anti-enteroviral activities of several anti-enteroviral compounds identified novel synergistic combinations against EV1 infection. The results are of high priority due to lack of effective drugs and schemes for treatment of enterovira; diseases.

The manuscript can be published after minor corrections.

Page 1 line 3 from the bottom. Please decipher the abbreviation BSA in the Introduction section.

Page 4 second paragraph, line 6. Change to read “drug interaction”

Page 4 second paragraph, line 2 from the bottom. Change to read “protected cells”

Author Response

Page 1 line 3 from the bottom. Please decipher the abbreviation BSA in the Introduction section.

Re: Corrected

Page 4 second paragraph, line 6. Change to read “drug interaction”

Re: Corrected

Page 4 second paragraph, line 2 from the bottom. Change to read “protected cells”

Re: Corrected. Many thanks.

Round 2

Reviewer 1 Report

Nil